# PREDICTION INCONSISTENCY HELPS GENERALIZABLE DETECTION OF ADVERSARIAL EXAMPLES

## ABSTRACT

A common way to defend models against adversarial examples (AEs) is to detect them based on their different properties from normal examples (NEs). However, current detection methods often suffer from poor generalization across model types or attack algorithms. In this work, we observe that an auxiliary model, with a different training strategy or architecture from the (target) primal model, tends to predict *differently* on the primal model's AEs but *similarly* on NEs. To this end, we propose Prediction Inconsistency Detection (PID), which simply leverages the above model prediction inconsistency, without training any detector. Experiments on CIFAR-10 and ImageNet demonstrate the superiority of our PID over 5 state-of-the-art detection methods. Specifically, PID achieves an improvement of 4.70%~8.44%, no matter whether the primal model is naturally or adversarially trained, and across 3 white-box, 3 black-box, and 1 mixed attack algorithms. We also show that using a naturally trained primal model and adversarially trained auxiliary model in PID yields a high AUC of 91.92% (84.43%) against strong, adaptive attacks on CIFAR-10 (ImageNet).

## 1 INTRODUCTION

Deep neural networks (DNNs) are vulnerable to adversarial examples (AEs) (Goodfellow et al., 2015; Carlini & Wagner, 2017), which consist of carefully crafted imperceptible adversarial perturbations and normal examples (NEs). AEs can mislead DNNs to output wrong predictions with high confidence regardless of original classification accuracy. As a common way of defense against AEs, detection methods aim to identify and reject AEs by leveraging their different properties from NEs (Xu et al., 2018; Monteiro et al., 2019; Tian et al., 2021; Zhang et al., 2023).

AE detection has been extensively studied, in both white-box (Ma et al., 2018; Monteiro et al., 2019; Tian et al., 2021; Zhang et al., 2023) and black-box (Xu et al., 2018; Tian et al., 2018) settings, based on whether the defended model is known to the detector. However, existing methods still suffer from poor generalization across model types or attack algorithms: (1) Most methods only consider naturally trained models, but turn out to perform poorly on adversarially trained models. This largely limits the potentially promising integration of AE detection and adversarial training techniques. (2) Most methods only consider white-box attacks, but turn out to perform poorly on other (unseen) black-box attacks (Aldahdooh et al., 2022).

To mitigate the above limitations, this paper proposes a black-box detection method exploiting the prediction inconsistency of models on AEs vs. NEs. Specifically, we observe a clear discrepancy in prediction results from the primary and auxiliary models on AEs, whereas their predictions remain consistent on NEs. When the auxiliary model differs from the primal model, either due to variations in training methods or differences in model architectures, it becomes more challenging for AEs to cross the decision boundary of the auxiliary model, as illustrated in Figure 1. As a result, the auxiliary model tends to assign low confidence scores to the labels predicted by the primal model. In contrast, both primal and auxiliary models typically maintain consistent predictions on NEs. Based on the above model prediction inconsistency, we design a simple yet flexible detection method named *Prediction Inconsistency Detection* (PID).

Specifically, we design four metrics for PID to measure the prediction inconsistency $I_{pred}$ between the primal and auxiliary models. Among these metrics, the most effective one works as follows: first, the primal model assigns a predicted hard label to the test sample; then, the confidence score on

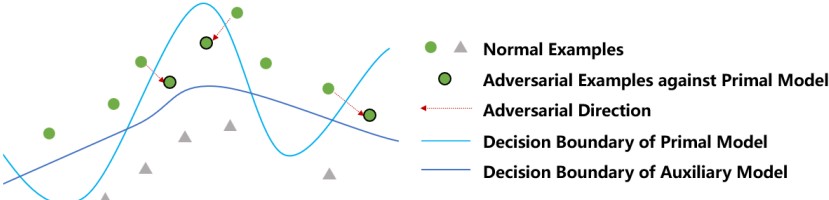

Figure 1: Adversarial examples against one (primal) model may not be adversarial against another (auxiliary) model, leading to prediction inconsistency that can be exploited for detection.

the same label from the auxiliary model is acquired, which is then employed to calculate prediction inconsistency $I_{pred}$. The higher the $I_{pred}$, the more likely the test sample is an AE. In addition to this new metric, our PID does not need to train a separate detector, as in existing methods that also rely on prediction inconsistency (Monteiro et al., 2019; Tian et al., 2021).

To summarize, our work makes the following contributions:

- We identify the significant prediction inconsistency between (the defended) primal model and a different auxiliary model on AEs vs. NEs, motivating our novel black-box detection method named PID without knowledge about the primal model and training any detector. In particular, we design a new metric for measuring prediction inconsistency based on the hard label from the primal model and that label's confidence score from the auxiliary model.

- We conduct extensive experiments on CIFAR-10 and ImageNet, which demonstrate the superior generalization of PID across different 2 model types (i.e., naturally and adversarially trained models) and 7 attack algorithms (i.e., 3 white-box, 3 black-box, and 1 mixed attacks), with an improvement of 4.70%~8.44%.

- We demonstrate that using adversarially trained primal and/or auxiliary models yields better robustness than using only naturally trained models. In particular, using a naturally trained primal model and adversarially trained auxiliary model yields a high AUC of 91.92% (84.43%) against strong, adaptive attacks on CIFAR-10 (ImageNet).

## 2  RELATED WORK

**Adversarial Attacks.** Adversarial attacks aim to mislead DNNs by adding imperceptible perturbations to NEs. Based on the attackers' knowledge, adversarial attacks can be roughly divided into two categories, i.e., white-box adversarial attacks and black-box adversarial attacks. White-box attacks assume full knowledge of the target model and include methods such as FGSM (Goodfellow et al., 2015), PGD (Madry et al., 2018), AutoAttack (AA) (Croce & Hein, 2020), C&W (Carlini & Wagner, 2017), and DeepFool (Moosavi-Dezfooli et al., 2016). Black-box attacks operate under limited knowledge and can be classified into decision-based, score-based, and transfer-based methods. Representative examples include Triangle Attack (TA) (Wang et al., 2022), Square (Andriushchenko et al., 2020), and VNI-FGSM (Wang & He, 2021).

**Detection of AEs.** In contrast to defense methods, which focus on enhancing robust test accuracy, detection methods identify and reject AEs by distinguishing them from NEs, thereby safeguarding the model. Most detection methods treat the protected model as a white-box model, generating AEs against the model to train the detector or extracting the features from the intermediate layers to analyze the differences between AEs and NEs. Monteiro et al. (2019) proposed the Bi-model Decision Mismatch Detector (BDMD), which uses the predicted soft labels from two models with different architectures on AEs as features to train a detector for AE identification. Similarly, Tian et al. (2021) introduced the Sensitivity Inconsistency Detector (SID), which trains a dual classifier with an additional Weighted Average Wavelet Transform layer. Combined with the primary classifier, the prediction inconsistency between the two classifiers is then used to train a detector to detect AEs. In Zhang et al. (2023), EPS-AD was developed, where the pre-trained diffusion model (Song et al., 2021) was adopted to estimate the expected perturbation score (EPS) of test samples, and EPS-based maximum mean discrepancy was used as the metric to measure the discrepancy between

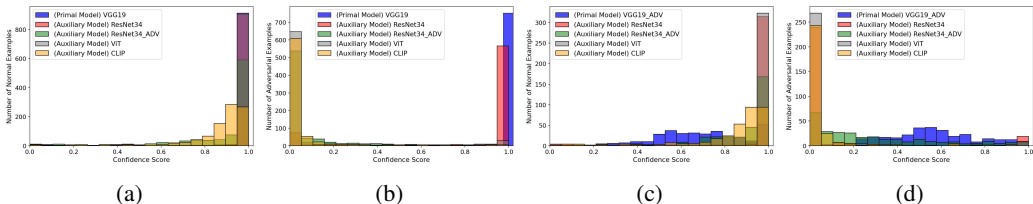

(a)           (b)           (c)           (d)

Figure 2: Confidence score distribution for *"Airplane"* label across primal and auxiliary models on (a) NEs classified by the naturally trained VGG19, (b) AEs classified by the naturally trained VGG19, (c) NEs classified by the adversarially trained VGG19, (d) AEs classified by the adversarially trained VGG19, respectively. It can be observed that an auxiliary model, differing from the primary model in training strategy or model architecture, tends to assign low confidence to the primary model's predictions on AEs, while preserving high confidence on NEs.

NEs and AEs. There are also some detection methods using the input transformation to disrupt the perturbations to cause the prediction fluctuation, which treats the protected model as a black-box model. Xu et al. (2018) proposed Feature Squeezing (FS), which reduces the perturbation space via several feature squeezers. Inputs showing significant prediction inconsistency before and after squeezing are flagged as adversarial.

## 3 METHOD

### 3.1 MOTIVATION

It is pointed out in Tian et al. (2021) that, AEs are sensitive to the fluctuation of the highly-curved region of the decision boundary, which can be exposed by training a dual classifier having dissimilar structures at the highly-curved regions with the original classifier while maintaining similar structures at the other regions. Inspired by this phenomenon, we explore whether AEs are sensitive to the fluctuation of decision boundaries caused by introducing auxiliary models, either with different training methods or in different model architectures compared to the primal model.

Specifically, we adopt the naturally trained VGG19 (denoted as VGG19) as the primal model on CIFAR-10 and select NEs that are correctly classified as *"Airplane"* by VGG19. Then four models are used as auxiliary models: the naturally trained ResNet34 (denoted as ResNet34), the adversarially trained ResNet34 (denoted as ResNet34_ADV), the naturally trained Vision Transformer (ViT)-L/16 (Dosovitskiy et al., 2021) (denoted as ViT), and the naturally trained Contrastive Language-Image Pre-Training (CLIP) model (Radford et al., 2021) (denoted as CLIP). Each auxiliary model is used to classify these NEs, and the confidence scores for the *"Airplane"* label serve as an intuitive indicator of sensitivity to decision boundary fluctuations, where lower scores indicate greater sensitivity. Furthermore, we choose AEs (generated by PGD attack with perturbation size of 8/255) that are wrongly classified as *"Airplane"* by VGG19 and use the same four auxiliary models to classify them and obtain the corresponding confidence scores for the *"Airplane"* label.

The confidence score distributions for *"Airplane"* label across primal and auxiliary models on NEs and AEs are depicted in Figure 2a and 2b, respectively. From these figures, we observe that for these NEs classified as *"Airplane"* by the primal model, all four auxiliary models output high confidence scores on the same label, with the CLIP model being slightly less confident. In contrast, for AEs labeled as *"Airplane"* by VGG19, ResNet34_ADV, ViT, and CLIP assign notably low confidence scores, while ResNet34 still gives high scores due to the transferability of AEs across Convolutional Neural Networks (CNNs).

Similar experiments are implemented using the adversarially trained VGG19 (denoted as VGG19_ADV) as the primal model on CIFAR-10. NEs and AEs predicted as *"Airplane"* by VGG19_ADV are selected, and their confidence score distributions across primal and auxiliary models are presented in Figure 2c and Figure 2d, respectively. It can be observed that four auxiliary models still assign low confidence scores for the AEs labeled as *"Airplane"* by VGG19_ADV. Furthermore, among the auxiliary models working with both naturally and adversarially trained models, ViT tend to assign the lowest confidence scores to the labels predicted by the primal model for AEs.

Based on the observations from our motivation experiments, we propose a detection method named *Prediction Inconsistency Detection* (PID), which can work with both naturally and adversarially trained models by introducing an auxiliary model and exploiting the model prediction inconsistency to tell NEs and AEs apart. The experiments above show that AEs are sensitive to decision boundary fluctuations caused by introducing auxiliary models, whether trained using different approaches (e.g., natural vs. adversarial training) or featuring distinct architectures (e.g., CNN vs. ViT). This sensitivity manifests as a noticeable drop in the confidence scores assigned by the auxiliary model to the hard labels predicted by the primary model. As illustrated in Figure 2, this prediction inconsistency is minimal for NEs but pronounced for AEs, making it an intuitive and effective signal for detection. The details of PID are given as follows.

### 3.2 DESIGN OF PID

**Preliminaries.** Let $f(\cdot)$ denote the primal model, which is the $k$-class classifier, and its output is hard label $y = \arg\max_i\{f_i(x)\}$, where $f_i(x) \in [0, 1]$ is the $i$-th class confidence score of the input $x$, and $i = 1, 2, \cdots, k$. Since our detection method is designed to work with both naturally and adversarially trained models, the primal model $f(\cdot)$ can also be obtained through adversarial training and its variants (Wong et al., 2020; Jia et al., 2024). $g(\cdot)$ represents the other $k$-class classifier used as the auxiliary model, and it outputs the confidence score $g(x) = \{g_1(x), g_2(x), \cdots, g_k(x)\}$, where $g_j(x) \in [0, 1]$ is the $j$-th class confidence score of the input $x$, and $j = 1, 2, \cdots, k$.

**Metrics for Quantifying Prediction Inconsistency.** The process of PID is illustrated in Figure 3, where 4 metrics are designed for measuring the prediction inconsistency.

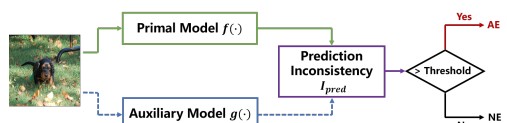

*Metric 1.* First, the test sample $x$ is input into the primal model $f(\cdot)$ and the assigned label $y$ can be acquired. Then $x$ is input into the auxiliary model $g(\cdot)$, and the confidence score $g_y(x)$ corresponding to label $y$ is obtained. Metric 1 for quantifying the prediction inconsistency $I_{pred}$ can be calculated as,

Figure 3: Overview of the proposed Prediction Inconsistency Detection (PID) for detecting AEs. The test sample are fed into primal and auxiliary models, respectively, to obtain the prediction inconsistency $I_{pred}$ by using the designed metrics. If $I_{pred}$ exceeds a threshold value, the test sample is judged to be adversarial.

$$y = \arg\max_i\{f_i(x)\},$$
$$I_{pred} = 1 - g_y(x). \tag{1}$$

If $x$ is an NE, $I_{pred}$ tends to be small (approaching 0), as a model trained on the same dataset is likely to classify it correctly with high confidence. Conversely, if $x$ is an AE, $I_{pred}$ can be large (approaching 1), especially when the attack fails to fool the auxiliary model $g(\cdot)$. Even when $x$ successfully misleads $g(\cdot)$, a low confidence score can still result in a high $I_{pred}$, making the AE more detectable.

Many detection methods (Xu et al., 2018; Tian et al., 2021) leverage differences between soft labels to estimate the likelihood of a test sample being adversarial. Following this idea, we propose metrics based on the soft label discrepancies between the primal and auxiliary models to quantify prediction inconsistency and detect AEs. For the rest of other metrics described in this subsection, both $k$-class classifiers $f(\cdot)$ and $g(\cdot)$ output confidence scores, where $f(x) = \{f_1(x), f_2(x), \cdots, f_k(x)\}$, $g(x) = \{g_1(x), g_2(x), \cdots, g_k(x)\}$.

*Metric 2.* We define Metric 2 as follows,

$$y = \arg\max_i\{f_i(x)\},$$
$$I_{pred} = f_y(x) - g_y(x), \tag{2}$$

where $f_y(x)$ and $g_y(x)$ denote the confidence scores for label $y$ from $f(\cdot)$ and $g(\cdot)$, respectively.

*Metric 3.* In Metric 3, prediction inconsistency is calculated using the $\ell_1$ norm of the difference between selected components of two prediction vectors. Specifically, we first sort confidence scores $f(x)$ in descending order and select the $n$ highest confidence scores $f(x)_{\text{top}-n} = \{f_{y_1}(x), f_{y_2}(x), \cdots, f_{y_n}(x)\}$ and corresponding labels $\{y_1, y_2, \cdots, y_n\}$. Then we obtain the confidence scores from $g(x)$ corresponding to labels $\{y_1, y_2, \cdots, y_n\}$, which are denoted as

$g(x)_{\text{top}-n} = \{g_{y_1}(x), g_{y_2}(x), \cdots, g_{y_n}(x)\}$, and the prediction inconsistency is represented by

$$I_{pred} = \|f(x)_{\text{top}-n} - g(x)_{\text{top}-n}\|_1. \tag{3}$$

*Metric 4.* In Metric 4, we use the $\ell_1$ norm of the difference between the entire prediction vectors $f(x)$ and $g(x)$ to measure the prediction inconsistency, which can be described as

$$I_{pred} = \|f(x) - g(x)\|_1. \tag{4}$$

A test sample is identified to be adversarial if the calculated $I_{pred}$ exceeds the pre-defined threshold. Note that although we evaluate all four metrics, Metric 1 is ultimately adopted for implementing the proposed PID, while the results for the remaining three are presented and discussed in Section 4.4.

**Choice of Auxiliary Models.** Previous studies have explored improving the transferability of AEs (Wang & He, 2021; Lin et al., 2019), showing that well-crafted AEs can often transfer between naturally trained CNN models. However, transferring AEs from a naturally trained CNN to an adversarially trained CNN remains challenging, which is also verified by our exploration shown in Figure 2b and 2d. This observation suggests that adversarially trained CNN models can serve as effective candidates for the auxiliary model in the proposed PID. Additionally, Mahmood et al. (2021) reported that AEs generated for CNNs struggle to fool ViT models (Dosovitskiy et al., 2021), potentially due to architectural differences. This makes ViTs another strong candidate for the auxiliary model, as they can effectively reveal prediction inconsistencies when facing AEs. Furthermore, pre-trained foundation models trained on large-scale datasets using self-supervised or weakly supervised learning can generalize across multiple downstream tasks (Awais et al., 2025), including the image classification task. Among them, the CLIP model (Radford et al., 2021), with its distinct architecture and outstanding zero-shot ability, can also be adopted as the auxiliary model to expose AEs.

Although several options are available for auxiliary models, as observed in Figure 2, ViT-L/16 exhibits the most significant prediction inconsistency from the primal model for AEs. As a result, we adopt ViT-L/16 as the default auxiliary model in our experiments to demonstrate the effectiveness of our black-box detection method. Results using adversarially trained CNNs and CLIP as auxiliary models are provided and analyzed in Section 4.4. Specifically, on CIFAR-10, we use adversarially trained ResNet34 (denoted as ResNet34_ADV) and CLIP-ViT-L/14, while on ImageNet, we employ adversarially pre-trained ConvNeXt-S (Liu et al., 2022) (denoted as ConvNeXt-S_ADV) and CLIP-ViT-L/14.

## 4 EVALUATION

### 4.1 EXPERIMENTAL SETTINGS

**Datasets and Models.** Two datasets, CIFAR-10 (Krizhevsky et al., 2009) and ImageNet (Deng et al., 2009), are adopted. For inference, we use the full test set of CIFAR-10, where images correctly classified by primal models are selected and attacked. On ImageNet, we randomly select 1000 correctly classified images (one per class) from the validation set for each primal model.

On CIFAR-10, we train the VGG19 model (Simonyan & Zisserman, 2015) under both natural and adversarial settings. On ImageNet, we use the ResNet50 model (He et al., 2016), and adopt pre-trained versions from both training settings. We adopt the ViT-L/16 model (Dosovitskiy et al., 2021) pre-trained on ImageNet as the auxiliary model in our PID. When evaluating on CIFAR-10, we fine-tune ViT-L/16 on the same dataset. Details of the primal and auxiliary models, as well as models used for implementing transfer-based black-box attacks, are provided in Section A.1.

**Attack Algorithms.** To comprehensively verify the effectiveness of PID, we adopt white-box, black-box, and mixed adversarial attacks. Particularly, to simulate a challenging adversarial environment, we (1) employ three types of black-box adversarial attacks, given their growing threat in real-world applications, and (2) vary attack parameters to generate perturbations of different magnitudes.

Table 1 summarizes the specific perturbation constraints for three types of attacks. Specifically, for PGD and AA, we set the $\ell_\infty$-constraint to $\epsilon = 1/255$ and $8/255$. For C&W, we choose $\kappa = 0$ and $1$, where $\kappa$ controls the confidence level of AEs. For VNI-FGSM, we set $\ell_\infty$-constraint to $\epsilon = 8/255$ on CIFAR-10 and $\epsilon = 16/255$ on ImageNet, respectively. More details of each implemented attack are given in Section A.2.

**Baseline Detections.** We compare our PID with 5 detection methods, namely FS (Xu et al., 2018), Diff-Pure (Nie et al., 2022), BDMD (Monteiro et al., 2019), SID (Tian et al., 2021), and EPS-AD (Zhang et al., 2023). Among them, FS and Diff-Pure are black-box detection methods, while BDMD, SID, and EPS-AD are white-box detection methods. Each method is combined with the naturally and adversarially trained models, respectively.

Table 1: Parameters of the implemented attack algorithms.

| Types | Attack | Norm | Constraint |
|---|---|---|---|
| White-box | PGD | $\ell_\infty$ | 1/255, 8/255 |
| White-box | C&W | $\ell_2$ | - |
| White-box | DeepFool | $\ell_2$ | - |
| Score-based black-box | Square | $\ell_\infty$ | 8/255 |
| Decision-based black-box | TA | $\ell_2$ | - |
| Transfer-based black-box | VNI-FGSM | $\ell_\infty$ | 8/255 or 16/255 |
| Mixed | AA | $\ell_\infty$ | 1/255, 8/255 |

FS and DiffPure serve as baselines on both the CIFAR-10 and ImageNet datasets. Notably, DiffPure is originally designed as a defense method, and we modify it as a black-box detection method considering the superior performance of the diffusion model on denoising images (Song et al., 2021; Ho et al., 2020). BDMD is evaluated on both CIFAR-10 and ImageNet, whereas SID is only applied to CIFAR-10, as training the dual classifier with the Weighted Average Wavelet Transform layer (Tian et al., 2021) on large-scale datasets like ImageNet is computationally expensive. Consistent with the original evaluation (Tian et al., 2021), we do not apply SID to ImageNet. Meanwhile, we compare PID with EPS-AD on ImageNet instead. Additionally, these three white-box detection methods will also be evaluated in a black-box detection scenario for a fair comparison with our black-box detection PID. We adjust the parameters of all detection methods to obtain their best performance. More detailed information are provided in Section A.3.

**Evaluation Metrics.** We use the AUC score, the Area Under the Receiver Operating Characteristic curve, to evaluate detection performance. This widely adopted metric provides an aggregate measure across all possible detection thresholds and serves as a unified index in prior works (Xu et al., 2018; Zhang et al., 2023; Tian et al., 2021; Ma et al., 2018). In addition, for each detection, we report the True Positive Rate (TPR) at a fixed False Positive Rate (FPR) of 5%.

## 4.2 EXPERIMENTAL RESULTS

**Results on CIFAR-10.** The evaluation of detection methods using AUC scores is summarized in Table 2, and TPRs at a fixed FPR can be found in Section A.4. As described in Section 3.2, our proposed PID is a black-box detection method, which utilizes the outputs of primal models without access to their internal outputs or parameters. For a fair comparison, BDMD and SID are first performed in a black-box way. To be specific, BDMD and SID are trained using the AEs against naturally and adversarially trained VGG16 models and evaluated on the AEs against naturally and adversarially trained VGG19 models, respectively. More details on implementation are provided in Section A.3. Meanwhile, the BDMD and SID in the white-box detection scenario are also implemented (denoted as BDMD* and SID* in Table 2, respectively).

As shown in Table 2, PID consistently achieves high AUC scores with both naturally and adversarially trained VGG19 models, reaching average scores of 99.29% and 99.30%, respectively. When working with the naturally trained primal model, DiffPure also achieves a good detection performance with an average AUC score of 94.59%, demonstrating its effectiveness on denoising AEs. In contrast, the performance of FS degrades dramatically when detecting strong attacks, which results from that strong AEs are actually more robust than NEs after being squeezed.

For BDMD, its performance drops significantly against Square and TA attacks. This can be attributed to that, when the primal model is naturally trained, strong attacks such as PGD can induce high-confidence misclassifications, whereas score-based and decision-based attacks typically stop once successful, resulting in lower confidence predictions. These differences result in the poor generalization of the BDMD across various attacks. For SID in the black-box scenario, its performance is limited, especially when detecting small-scale AEs such as PGD-1/255 and AA-1/255. After performing SID in the white-box scenario, where the detector is trained and tested on AEs generated by the same attack against the same primal model, its performance improves, reaching an average AUC score of 87.58%. Nevertheless, this score remains constrained.

When combined with adversarially trained models, FS, DiffPure, and SID show a decline in performance, with average AUC scores falling below 80%, whereas BDMD achieves an AUC of 90.86%.

Table 2: Comparison of AUC scores (%) of detecting AEs on CIFAR-10, where NAT (ADV) means the primal model is naturally (adversarially) trained. BDMD* and SID* are not directly comparable to others since they are in the ideal, white-box scenario.

| Primal Model | Detection Method | PGD $\epsilon=\frac{1}{255}$ | PGD $\epsilon=\frac{8}{255}$ | AA $\epsilon=\frac{1}{255}$ | AA $\epsilon=\frac{8}{255}$ | C&W $\kappa=0$ | C&W $\kappa=1$ | DeepFool | Square | TA | VNI-FGSM | Average |
|---|---|---|---|---|---|---|---|---|---|---|---|---|
| NAT | FS | 83.84 | 56.76 | 87.24 | 63.11 | 93.58 | 89.47 | 92.15 | 92.96 | 93.10 | 71.42 | 82.36 |
|  | DiffPure | 88.93 | 97.75 | 91.57 | 97.90 | 95.09 | 95.00 | 97.02 | 92.91 | 92.80 | 96.92 | 94.59 |
|  | BDMD | 88.05 | 98.48 | 89.57 | 98.66 | 66.10 | 71.38 | 85.60 | 22.58 | 5.88 | 95.12 | 72.14 |
|  | BDMD* | 91.53 | 98.09 | 91.81 | 98.75 | 75.87 | 79.01 | 89.10 | 35.41 | 20.71 | 95.87 | 77.62 |
|  | SID | 67.83 | 92.84 | 70.59 | 90.52 | 79.29 | 79.30 | 83.14 | 89.31 | 95.36 | 73.03 | 82.12 |
|  | SID* | 68.09 | 99.35 | 72.23 | 99.23 | 85.54 | 86.21 | 86.83 | 94.98 | 97.65 | 85.65 | 87.58 |
|  | PID (Ours) | **99.81** | **98.54** | **99.85** | **98.71** | **99.93** | **99.45** | **99.86** | **99.88** | **99.85** | **97.02** | **99.29** |
| ADV | FS | 49.40 | 58.22 | 49.29 | 61.67 | 81.79 | 84.22 | 70.81 | 55.29 | 76.40 | 50.88 | 63.80 |
|  | DiffPure | 50.81 | 74.55 | 50.77 | 76.38 | 87.00 | 88.55 | 83.77 | 63.60 | 84.38 | 52.77 | 71.26 |
|  | BDMD | 81.93 | 98.02 | 77.11 | 97.83 | 94.00 | 97.55 | 95.00 | 92.72 | 95.76 | 78.71 | 90.86 |
|  | BDMD* | 88.54 | 98.25 | 82.05 | 98.20 | 94.68 | 97.34 | 95.29 | 94.11 | 95.88 | 80.98 | 92.53 |
|  | SID | 51.96 | 75.90 | 40.09 | 78.08 | 89.92 | 87.07 | 89.64 | 82.80 | 90.42 | 52.49 | 73.84 |
|  | SID* | 63.15 | 75.69 | 51.67 | 76.25 | 90.10 | 89.27 | 92.28 | 80.80 | 93.52 | 62.88 | 77.56 |
|  | PID (Ours) | **99.87** | **99.68** | **99.88** | **99.69** | **99.91** | **99.90** | **99.55** | **99.82** | **99.82** | **94.90** | **99.30** |

Table 3: Comparison of AUC scores (%) of detecting AEs on ImageNet, where NAT (ADV) means the primal model is naturally (adversarially) trained. BDMD* and EPS-AD* are not directly comparable to others since they are in the ideal, white-box scenario.

| Primal Model | Detection Method | PGD $\epsilon=\frac{1}{255}$ | PGD $\epsilon=\frac{8}{255}$ | AA $\epsilon=\frac{1}{255}$ | AA $\epsilon=\frac{8}{255}$ | C&W $\kappa=0$ | C&W $\kappa=1$ | DeepFool | Square | TA | VNI-FGSM | Average |
|---|---|---|---|---|---|---|---|---|---|---|---|---|
| NAT | FS | 91.50 | 24.48 | 90.07 | 32.08 | 92.46 | 84.79 | 95.43 | 87.64 | 88.17 | 74.75 | 76.14 |
|  | DiffPure | 96.38 | 97.59 | 97.70 | 97.97 | 91.53 | 95.42 | 86.55 | 85.22 | 83.62 | 96.30 | 92.83 |
|  | BDMD | 93.89 | 93.86 | 94.45 | 93.88 | 93.32 | 93.65 | 92.07 | 91.20 | 90.92 | 92.59 | 92.98 |
|  | BDMD* | 93.04 | 93.46 | 93.76 | 93.34 | 92.51 | 93.07 | 90.90 | 89.47 | 89.94 | 92.43 | 92.19 |
|  | EPS-AD | 96.64 | **99.89** | 96.90 | **99.77** | 92.07 | **99.84** | 55.48 | 62.43 | 55.55 | **99.38** | 85.80 |
|  | EPS-AD* | 99.33 | 99.91 | 99.55 | 99.88 | 94.85 | 99.95 | 57.55 | 65.24 | 58.68 | 99.95 | 87.49 |
|  | PID (Ours) | **98.45** | 98.54 | **98.74** | 98.90 | **98.17** | 98.20 | **98.36** | **97.80** | **97.80** | 98.11 | **98.31** |
| ADV | FS | 48.86 | 72.00 | 49.04 | 75.59 | 72.07 | 76.89 | 74.75 | 59.99 | 81.39 | 62.77 | 67.34 |
|  | DiffPure | 47.33 | 82.71 | 50.26 | 86.91 | 56.96 | 72.16 | 78.09 | 58.32 | 79.41 | 62.18 | 67.43 |
|  | BDMD | 61.74 | 81.87 | 61.78 | 81.97 | 68.35 | 79.04 | 78.53 | 75.39 | 81.91 | 68.70 | 73.93 |
|  | BDMD* | 58.73 | 81.94 | 59.98 | 81.74 | 67.73 | 78.18 | 79.31 | 73.00 | 81.78 | 68.93 | 73.13 |
|  | EPS-AD | 87.56 | **99.83** | 85.48 | **99.89** | 96.58 | **99.82** | 94.36 | 66.03 | 85.76 | **100.00** | 91.53 |
|  | EPS-AD* | 95.20 | 99.12 | 95.27 | 99.14 | 97.78 | 99.65 | 95.52 | 65.35 | 88.60 | 100.00 | 93.56 |
|  | PID (Ours) | **95.98** | 96.17 | **98.20** | 97.83 | **97.23** | 97.11 | **98.06** | **96.90** | **98.34** | 92.23 | **96.81** |

In contrast, our PID still reaches the highest AUC score, exhibiting its good generalization and flexibility. This performance gap can be explained by two key factors. First, adversarial training makes the model less overconfident (Grabinski et al., 2022), leading to less noticeable differences in predictions before and after image transformations or denoising, which weakens methods that rely on such differences. Second, the increased robustness of adversarially trained models leads to fewer successful AEs, resulting in insufficient training samples for the detector.

**Results on ImageNet.** The evaluation of detection methods using AUC scores is shown in Table 3, and TPRs at a fixed FPR can be found in Section A.4. In the black-box detection scenario, BDMD and EPS-AD are trained using the AEs against naturally and adversarially trained ResNet101 models and evaluated on the AEs against naturally and adversarially trained ResNet50 models, respectively. Further implementation details can be found in Section A.3. For the white-box detection scenario, BDMD and EPS-AD are also implemented (denoted as BDMD* and EPS-AD* in Table 3). It is worth noting that this setup is relatively mild, as the primal models employed in the training and evaluation of the detector share very similar architectures.

Table 4: Detection performance of AUC scores (%) on PGD vs. an adaptive attack.

| Dataset | Primal Model | Auxiliary Model | PGD | Adaptive |
|---------|--------------|-----------------|-----|----------|
| CIFAR-10 | NAT | ResNet34_ADV | 96.56 | 91.92 |
| | | ViT-L/16 | 98.54 | 72.20 |
| | | CLIP-ViT-L/14 | 97.55 | 33.44 |
| | ADV | ResNet34_ADV | 91.46 | 76.43 |
| | | ViT-L/16 | 99.68 | 51.82 |
| | | CLIP-ViT-L/14 | 98.75 | 16.27 |
| ImageNet | NAT | ConvNeXt-S_ADV | 97.53 | 84.43 |
| | | ViT-L/16 | 98.54 | 27.39 |
| | | CLIP-ViT-L/14 | 95.91 | 16.18 |
| | ADV | ConvNeXt-S_ADV | 97.26 | 65.93 |
| | | ViT-L/16 | 96.17 | 23.60 |
| | | CLIP-ViT-L/14 | 96.13 | 8.04 |

Table 5: Detection performance of PID by employing different metrics for quantifying prediction inconsistency on CIFAR-10. The average AUC scores (%) over 10 attacks are shown, with full results available in Section A.7.

| Metric | NAT | ADV |
|--------|-----|-----|
| $I_{pred} = -g_y(x)$ | **99.29** | **99.30** |
| $I_{pred} = f_y(x) - g_y(x)$ | 98.58 | 96.74 |
| $I_{pred} = \|f(x)_{\text{top}-n} - g(x)_{\text{top}-n}\|_1$ | 98.75 | 77.76 |
| $I_{pred} = \|f(x) - g(x)\|_1$ | 98.96 | 78.94 |

It can be seen from Table 3 that, our PID still outperforms other detection methods when combined with both naturally and adversarially trained models. Experimental results in Table 3 is obtained on the randomly sampled 1000 images (one from each class). To mitigate the effect of randomness, we additionally sample two separate sets of 1000 images each, and the experimental results can be found in Section A.5. In contrast, the detection performance of FS, DiffPure, and BDMD exhibits a noticeable decrease after working with the adversarially trained model. While EPS-AD exhibits strong performance in detecting AEs across multiple attack types, its effectiveness drops significantly when faced with score-based and decision-based black-box attacks, i.e., Square and TA attacks, either combined with naturally or adversarially trained models. This also emphasizes the necessity of introducing black-box attacks in the reliable evaluation of detection methods.

### 4.3 ADAPTIVE ATTACKS

We now evaluate the robustness of our PID under the adaptive attack, where the attacker has access to the full knowledge of both primal and auxiliary models and aims to mislead both models simultaneously. Since Metric 1 is used in PID, i.e., $I_{pred} = 1 - g_y(x)$, an intuitive adaptive attack strategy is to maximize $g_y(x)$. To achieve this, we design three adaptive attacks. The strongest attack is presented below, while detailed information on the other two is provided in Section A.6.

To be specific, we (1) perform an untargeted attack on $f(\cdot)$ to force it to predict a wrong label, and (2) jointly attack both $f(\cdot)$ and $g(\cdot)$ to make $g(\cdot)$ misclassify the AE into the same wrong label. This process can be formulated as follows,

$$\text{Find } r_1 \text{ s.t. } t = \arg\max_i f_i(x + r_1), \quad t \neq y_{\text{true}}, \quad \|r_1\|_\infty \leq \epsilon, \tag{5}$$

$$r_2 = \arg\min_{\|r_1 + r_2\|_\infty \leq \epsilon} \left[ \mathcal{L}\big(f(x + r_1 + r_2), t\big) + \lambda \mathcal{L}\big(g(x + r_1 + r_2), t\big) \right], \tag{6}$$

where $y_{true}$ is the ground-truth label of $x$, $\mathcal{L}(\cdot, \cdot)$ is the loss function, $\epsilon$ is the perturbation constraint, and $\lambda$ is a trade-off parameter. We use the PGD attack strategy to find perturbations $r_1$ and $r_2$, where $r_1$ is obtained by increasing the loss $\mathcal{L}\big(f(x + r_1), y_{\text{true}}\big)$, and $r_2$ is obtained following Eq.6 after $r_1$ cause the misclassification. To ensure strong attack performance, the perturbation constraint is set to $\epsilon = 8/255$, with total 100 iterations and a step size $1/255$. Furthermore, we set $\lambda = 99$ to encourage $g(\cdot)$ assign high confidence to the wrong label $t$.

Table 4 reports the performance of PID against the two-phase adaptive attack on CIFAR-10 and ImageNet with three kinds of auxiliary models. PID maintains relatively strong detection performance with adversarially trained CNNs, benefiting from the robustness of adversarial training (AT) and the distinct decision boundaries between differently trained models. However, PID shows the apparent performance drop when the auxiliary models are ViT or CLIP, likely due to their vulnerability to adversarial threats (Mahmood et al., 2021; Zhang et al., 2022), which enables the adaptive attack to jointly fool both primal and auxiliary models.

It is worth noting that adaptive attacks represent an idealized threat model where the attacker has full knowledge of both the classifier and the detector, under which many detection methods have been shown to fail (Tramer et al., 2020; Tramer, 2022). Since AT is widely regarded as a strong defense, PID's ability to incorporate AT ensures that future improvements in AT can be directly leveraged, further enhancing its robustness against adaptive attacks.

## 4.4 ABLATION STUDY

**Metrics for Quantifying Prediction Inconsistency.** Here, we discuss how the metrics for quantifying prediction inconsistency impact the effectiveness of the PID. Four metrics described in Section 3.2 are employed to measure prediction inconsistency for implementing PID, where we use $n = 3$ in Metric 3. The experimental results on CIFAR-10 are summarized in Table 5.

It can be observed that four metrics yield similar results when the protected primal model is naturally trained, yet Metric 1 remains the best. However, after the primal model is adversarially trained, Metric 3 and 4 lead to a performance drop, whereas Metric 1 still enables PID to achieve strong detection performance. This drop occurs because adversarial training reduces the primal model's overconfidence on both NEs and AEs, causing the differences between confidence scores from $f(x)$ and $g(x)$ to increase for NEs and decrease for AEs, particularly for weak AEs with low confidence on incorrectly predicted labels. The reduced disparity makes it more difficult for other metrics to distinguish NEs from AEs. In contrast, Metric 1 relies only on the hard label from $f(x)$, avoiding the fluctuations in confidence scores introduced by the training strategy, and thus allowing PID to remain consistently effective.

**Choice of Auxiliary Model.** As discussed in Section 3.2, in addition to ViT, the adversarially trained CNN and CLIP model can also serve as the auxiliary model in our PID. The corresponding experimental results for PID using these three kinds of auxiliary models on CIFAR-10 and ImageNet are summarized in Table 6.

Across both datasets, PID remains stable and effective in detecting AEs, regardless of the auxiliary model used. Specifically, ViT-L/16 consistently achieves the highest AUC scores with both naturally or adversarially trained models, which can be attributed to its high clean accuracy and significant architecture difference from CNNs. Although the detection performance of the adversarially trained CNN and CLIP models varies across datasets, their average AUC scores still surpass those of the other five baselines shown in Table 2 and 3, confirming the effectiveness and the flexibility of the proposed PID.

Table 6: Detection performance of PID by employing different models as the auxiliary model on CIFAR-10 and ImageNet, where NAT (ADV) means the primal model is naturally (adversarially) trained. The average AUC scores (%) over 10 attacks are shown, with full results available in Section A.8.

| Dataset | Auxiliary Model | NAT | ADV |
|---|---|---|---|
| CIFAR-10 | ResNet34_ADV | 95.14 | 92.75 |
| | ViT-L/16 | **99.29** | **99.30** |
| | CLIP-ViT-L/14 | 98.36 | 98.37 |
| ImageNet | ConvNeXt-S_ADV | 96.93 | 95.06 |
| | ViT-L/16 | **98.31** | **96.81** |
| | CLIP-ViT-L/14 | 95.34 | 94.28 |

The computational costs of PID with the three auxiliary models, compared to other baseline detections, are reported in Section A.9, demonstrating that PID remains lightweight even when using large-scale auxiliary models. Furthermore, Section A.10 explores a broader range of auxiliary models, with results confirming that PID generalizes well across diverse auxiliary architectures.

## 5 CONCLUSION

This paper proposes a lightweight detection method named PID, which leverages the prediction inconsistency between the primal and auxiliary models to detect AEs without requiring any prior model-specific knowledge or training a separate detector. Our method demonstrates strong generalization, as it not only maintains compatibility with both naturally and adversarially trained models, but also achieves consistently high detection performance across a comprehensive evaluation, including white-box, black-box, and mixed attacks with varying strengths on two widely used datasets.

## ETHICS STATEMENT

This work develops a lightweight method for detecting adversarial examples, which does not involve potential violations of the Code of Ethics. Authors read the Code of Ethics, adhere to it, and explicitly acknowledge this during the submission process.

## REPRODUCIBILITY STATEMENT

The detailed information about the implementation of the proposed PID is provided in Section 3.2 and Section 4.1. The data used in this paper is open-source, and the selection process is stated in Section 4.1. Experimental settings are stated in Section 4.1, and further details are provided in Section A.1 to Section A.3 in Appendix. The code is included in the supplementary material, along with sufficient instructions to faithfully reproduce the main experimental results.

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

## A  APPENDIX

### A.1  DETAILS ON EMPLOYED MODELS

**Primal Models.** On CIFAR-10, we adopt VGG19 as the primal model. Under natural training, the model is trained for 150 epochs with a batch size of 128 using the SGD optimizer, with a weight decay of $5 \times 10^{-4}$. The initial learning rate is 0.1, which is reduced by a factor of 0.1 at the 50th and 100th epochs. It achieves 92.21% accuracy on the CIFAR-10 test set. For adversarial training, we follow Madry et al. (2018) and use PGD with an $\ell_\infty$ perturbation bound of 8/255, step size of 2/255, and 10 iterations. The model is adversarially trained for 200 epochs with a batch size of 128, starting from an initial learning rate of 0.005, decayed by 0.1 at the 50th and 100th epochs. It achieves 74.19% accuracy on the CIFAR-10 test set. On ImageNet, we adopt ResNet50 as the primal model and adopt pre-trained weights under both natural[1] and adversarial[2] training settings.

**Auxiliary Models.** In this work, the employed auxiliary models fall into three categories: adversarially trained CNNs, ViTs, and CLIP models. For adversarially trained CNNs, we adopt the adversarially trained ResNet34 (denoted as ResNet34_ADV) on CIFAR-10, which is trained under the same setting as the adversarially trained VGG19. It achieves 83.44% accuracy on the CIFAR-10 test set. On ImageNet, we adopt the pre-trained ConvNeXt-S under the adversarial training settings[2] (denoted as ConvNeXt-S_ADV). For ViTs, we use the pre-trained ViT-L/16[3] on ImageNet. On CIFAR-10, we fine-tune it for 20 epochs with a learning rate of $1 \times 10^{-5}$, achieving 98.35% test accuracy. For CLIP models, we use the pre-trained CLIP-ViT-L/14[4] on both CIFAR-10 and ImageNet.

**Models in Transfer-based Black-box Attacks.** To implement the VNI-FGSM attack on both CIFAR-10 and ImageNet, we train substitute models with architectures similar to the corresponding primal models to enhance adversarial transferability. On CIFAR-10, we use the naturally trained VGG13, which is trained under the same setting as the naturally trained VGG19, achieving 92.84% test accuracy. On ImageNet, we adopt the pre-trained ResNet152[1].

### A.2  DETAILS ON IMPLEMENTED ATTACKS

**White-box attacks.** The $\ell_\infty$-constraints of **PGD** attack are set to 1/255 and 8/255. The former uses a step size of 1/255 with 2 iterations, while the latter uses a step size of 2/255 with 10 iterations. For **C&W** attack, the hyper-parameter $\kappa$ that controls the confidence level of the generated AE is set to be 0 and 1 to generate AEs with different strengths. For **DeepFool** attack, the maximum numbers of iterations are 30 and 500 on CIFAR-10 and ImageNet, respectively, and the constant used to enlarge the last step to cross over the decision boundary is 0.02.

**Black-box attacks.** The constraint of **Square** attack is set to 8/255, and the query limits are set as follows: 5000 for the naturally trained model on CIFAR-10, 10,000 for the adversarially trained model on CIFAR-10, 10,000 for the naturally trained model on ImageNet, and 20,000 for the adversarially trained model on ImageNet. Following the implementation of **TA** in Wang et al. (2022), the maximum number of iterations in each subspace $N = 2$, and the dimension of directional line $d = 3$. For updating angle $\alpha$, the change rate $\gamma = 0.01$, the constant $\lambda = 0.05$, and the parameter restricting the upper and lower bounds $\tau = 0.1$ are used. The query numbers are set to be 500 and 1000 on CIFAR-10 and ImageNet, respectively. For the transfer-based **VNI-FGSM** attack, VGG13 and ResNet152 models are used as substitute models on CIFAR-10 and ImageNet, respectively. Following Wang & He (2021), the number of sampled examples in the neighborhood $N = 20$, the upper bound of neighborhood $\beta = 1.5$, and the decay factor $\mu = 1.0$. The perturbation sizes are set to $\epsilon = 8/255$ and $\epsilon = 16/255$ on CIFAR-10 and ImageNet, respectively, the number of iteration $n = 10$, and step size $\alpha = \epsilon/n$.

**Mixed attacks. AA** is an ensemble attack consisting of the untargeted APGD-CE, targeted APGD-DLR, targeted FAB, and the untargeted Square attack with 5000 query times. The $\ell_\infty$-constraints of

---

[1] https://github.com/pytorch/vision/blob/main/torchvision/models/resnet.py
[2] https://github.com/thu-ml/ares/tree/main/robust_training
[3] https://github.com/pytorch/vision/blob/main/torchvision/models/vision_transformer.py
[4] https://huggingface.co/openai/clip-vit-large-patch14

Table 7: ASR (%) of each attack on CIFAR-10 and ImageNet, where NAT (ADV) means the primal model is naturally (adversarially) trained.

| Dataset | Primal Model | PGD $\epsilon = \frac{1}{255}$ | PGD $\epsilon = \frac{8}{255}$ | AA $\epsilon = \frac{1}{255}$ | AA $\epsilon = \frac{8}{255}$ | C&W $\kappa = 0$ | C&W $\kappa = 1$ | DeepFool | Square | TA | VNI-FGSM |
|---------|------|------|-------|------|-------|-------|-------|--------|-------|-------|-------|
| CIFAR-10 | NAT | 38.21 | 100.00 | 46.31 | 100.00 | 100.00 | 100.00 | 100.00 | 99.87 | 99.80 | 89.35 |
| | ADV | 4.92 | 45.37 | 6.36 | 50.51 | 62.37 | 77.92 | 100.00 | 44.68 | 99.73 | 3.48 |
| ImageNet | NAT | 95.20 | 100.00 | 100.00 | 100.00 | 100.00 | 100.00 | 100.00 | 100.00 | 99.70 | 97.70 |
| | ADV | 10.50 | 75.50 | 13.30 | 81.50 | 30.90 | 70.70 | 100.00 | 62.00 | 99.40 | 9.24 |

AA are set to be 1/255 and 8/255 for each model on CIFAR-10 and ImageNet. When implementing AA against the adversarially trained model on ImageNet, untargeted versions of APGD-DLR and FAB attacks are used instead, due to high computational cost.

Except for TA, which is implemented using the code[5] provided by authors, all other adversarial attacks are conducted using torchattacks (Kim, 2020), and the attack success rate (ASR) of each attack is shown in Table 7. It can be observed from Table 7 that adversarially trained models are still vulnerable to black-box attacks, and this phenomenon has also been observed and demonstrated in Dong et al. (2020). In addition, the performance of some detection methods can be distorted by black-box attacks (Aldahdooh et al., 2022). As a result, it is crucial to evaluate both detection and defense methods using the black-box attack scenario. Moreover, a detection-based module that complements defended models needs further exploration.

### A.3 DETAILS ON DETECTION BASELINES

Five detection methods are employed in our evaluation for comparison, and their implementation details are introduced as follows. We adjust the parameters of these detection baselines to obtain the best detection performance.

**Feature Squeeze (FS) (Black-Box Detection).** Bit depth squeezer, local smoothing squeezer, and non-local smoothing squeezer are adopted in our work, consistent with those used in Xu et al. (2018), and the implementation follows the released code[6]. We reduce the original 8-bit images in test sets of two datasets to 5-bit images. Local smoothing squeezer is the median smoothing method with the $2 \times 2$ sliding window, where the center pixel is located at the lower right, and *reflect padding* is used for pixels on the edge. These two squeezers remain invariant when implementing FS. For the non-local smoothing squeezer, a variant of the Gaussian kernel is used. In different settings, the following parameters are applied:

- For the naturally trained VGG19 on CIFAR-10, we set the search window size $a = 13$, patch size $b = 3$, and filter strength $c = 2$.
- For the adversarially trained VGG19 on CIFAR-10, the parameters are $a = 13$, $b = 3$, and $c = 4$.
- For the naturally trained ResNet50 on ImageNet, the parameters are $a = 11$, $b = 3$, and $c = 4$.
- For the adversarially trained ResNet50 on ImageNet, we use $a = 11$, $b = 2$, and $c = 3$.

The calculation of the probability of a test sample being an AE strictly follows Xu et al. (2018).

**DiffPure (Black-Box Detection).** DiffPure is originally designed as a defense method, and we modify it as a black-box detection method in this work. We use the strategies described in Nie et al. (2022) to purify the inputs, following the released code[7], where the pre-trained diffusion models Score-SDE (Song et al., 2021) and Guided Diffusion (Dhariwal & Nichol, 2021) on CIFAR-10 and ImageNet are adopted, respectively. On CIFAR-10, timesteps $t^* = 0.10$ and $t^* = 0.15$ are used for the naturally and adversarially trained VGG19 models, respectively. On ImageNet, $t^* = 0.15$ and $t^* = 0.30$ are employed for the naturally and adversarially trained ResNet50 models, respectively.

---

[5]https://github.com/xiaosen-wang/TA
[6]https://github.com/mzweilin/EvadeML-Zoo
[7]https://github.com/NVlabs/DiffPure

Table 8: Comparison of TPRs (%) of detecting AEs on CIFAR-10, where FPR is fixed at 5%, and NAT (ADV) means the primal model is naturally (adversarially) trained. BDMD* and SID* are not directly comparable to others since they are in the ideal, white-box scenario.

| Primal Model | Detection Method | PGD $\epsilon = \frac{1}{255}$ | PGD $\epsilon = \frac{8}{255}$ | AA $\epsilon = \frac{1}{255}$ | AA $\epsilon = \frac{8}{255}$ | C&W $\kappa = 0$ | C&W $\kappa = 1$ | DeepFool | Square | TA | VNI-FGSM | Average |
|---|---|---|---|---|---|---|---|---|---|---|---|---|
| NAT | FS | 31.62 | 14.28 | 41.17 | 21.11 | 48.28 | 44.91 | 53.68 | 12.05 | 7.27 | 27.75 | 30.21 |
| | DiffPure | 39.40 | 89.60 | 49.53 | 90.54 | 49.02 | 52.47 | 76.76 | 8.02 | 4.05 | 87.13 | 54.65 |
| | BDMD | 87.60 | **95.90** | 89.32 | **96.66** | 65.65 | 70.17 | 85.50 | 21.09 | 4.53 | 88.96 | 70.54 |
| | BDMD* | 90.89 | 95.53 | 91.59 | 96.79 | 75.10 | 77.56 | 88.91 | 33.05 | 17.99 | 89.90 | 75.73 |
| | SID | 27.29 | 60.30 | 26.66 | 48.99 | 42.19 | 38.69 | 83.14 | 51.95 | 79.83 | 20.53 | 47.96 |
| | SID* | 22.38 | 97.29 | 36.94 | 96.50 | 62.79 | 55.35 | 61.27 | 82.53 | 91.49 | 48.20 | 65.47 |
| | PID | **99.35** | 94.72 | **99.53** | 95.30 | **99.78** | **98.03** | **99.58** | **99.58** | **99.50** | **89.11** | **97.45** |
| ADV | FS | 3.84 | 7.01 | 4.24 | 9.34 | 33.09 | 40.81 | 11.71 | 5.07 | 19.49 | 2.32 | 13.69 |
| | DiffPure | 9.04 | 30.48 | 6.99 | 34.77 | 55.64 | 61.03 | 39.45 | 17.83 | 41.38 | 7.72 | 30.43 |
| | BDMD | 29.59 | 93.46 | 32.84 | 91.97 | 77.54 | 92.51 | 86.25 | 55.38 | 89.12 | 40.54 | 68.92 |
| | BDMD* | 55.89 | 95.13 | 48.52 | 95.33 | 89.54 | 95.62 | 89.69 | 81.30 | 91.62 | 51.35 | 79.40 |
| | SID | 4.55 | 34.95 | 2.08 | 36.94 | 70.34 | 55.82 | 67.89 | 53.01 | 71.74 | 12.66 | 41.00 |
| | SID* | 41.82 | 29.22 | 20.83 | 34.19 | 62.85 | 66.42 | 74.00 | 52.91 | 75.75 | 11.39 | 46.94 |
| | PID | **99.45** | **98.72** | **99.36** | **98.80** | **99.83** | **99.81** | **98.29** | **99.37** | **99.36** | **79.54** | **97.25** |

After obtaining the purified test sample $x'$ for the test sample $x$, we adopt the $\ell_1$ norm of difference of whole prediction vectors from the primal model on these two samples to calculate the probability of the test sample being an AE (denoted as $prob$), which can be described as,

$$prob = \|f(x) - f(x')\|_1, \tag{7}$$

where the primal model outputs confidence scores $f(x) = \{f_1(x), f_2(x), \cdots, f_k(x)\}$, $f_i(x) \in [0, 1]$, and $k$ is the number of classes.

**Bi-model Decision Mismatch Detector (BDMD) (White-Box Detection).** BDMD (Monteiro et al., 2019) leverages the predicted soft labels from two models with different architectures on AEs to train a detector, implemented as a Support Vector Machine (SVM) with RBF kernel. To make BDMD a competitive baseline, we adopt ViT-L/16 (the optimal auxiliary model in PID) as the second model in addition to the attacked primal model.

For the white-box setting (denoted as BDMD*), to evaluate the generalization of BDMD across different attacks, the detector is trained on AEs generated by PGD-8/255. On CIFAR-10, PGD-8/255 AEs generated from the entire test set are split into 60% for training and 40% for testing, while all AEs are used when detecting other types of attacks. On ImageNet, we additionally sample 5000 clean images to generate PGD-8/255 AEs for training and evaluate the detector on all types of attacks.

For the black-box setting, we use naturally and adversarially trained VGG16 models as primal models on CIFAR-10, and ResNet101 models on ImageNet. The naturally trained VGG16 achieves 92.19% accuracy, while the adversarially trained one (PGD-8/255) achieves 79.29%. For ResNet101, we use pre-trained weights under natural[1] and adversarial[2] training. On CIFAR-10, detectors are trained on PGD-8/255 AEs from naturally and adversarially trained VGG16 models, with the same 60%-40% split, and tested on all AEs from naturally and adversarially trained VGG19 models. On ImageNet, detectors are trained on PGD-8/255 AEs from ResNet101 and tested on AEs from ResNet50, with 5000 additional clean images used to generate training AEs.

**Sensitivity Inconsistency Detector (SID) (White-Box Detection).** SID (Tian et al., 2021) trains a dual classifier with the transformed decision boundary by adding the Weighted Average Wavelet Transform (WAWT) layer. The prediction differences between the primal classifier and dual classifier are used to train a detector composed of two fully connected layers. The implementation of SID can be computationally expensive on ImageNet, so we adopt it only on CIFAR-10.

We train a dual classifier for naturally and adversarially trained VGG19 models following Tian et al. (2021) and the released code[8], where the dual classifier consists of the same VGG19 architecture and a WAWT layer. For the white-box SID setting (denoted as SID*), we evaluate detection performance

[8]https://github.com/JinyuTian/SID

Table 9: Comparison of TPRs (%) of detecting AEs on ImageNet, where FPR is fixed at 5%, and NAT (ADV) means the primal model is naturally (adversarially) trained. BDMD* and EPS-AD* are not directly comparable to others since they are in the ideal, white-box scenario.

| Primal Model | Detection Method | PGD $\epsilon = \frac{1}{255}$ | PGD $\epsilon = \frac{8}{255}$ | AA $\epsilon = \frac{1}{255}$ | AA $\epsilon = \frac{8}{255}$ | C&W $\kappa = 0$ | C&W $\kappa = 1$ | DeepFool | Square | TA | VNI-FGSM | Average |
|---|---|---|---|---|---|---|---|---|---|---|---|---|
| NAT | FS | 69.75 | 4.60 | 68.80 | 8.00 | 63.60 | 45.40 | 78.40 | 28.00 | 26.18 | 24.69 | 41.74 |
| | DiffPure | 80.78 | 86.50 | 87.40 | 89.10 | 50.20 | 70.20 | 6.70 | 13.10 | 4.81 | 82.72 | 57.15 |
| | BDMD | 22.37 | 20.50 | 22.80 | 17.60 | 21.90 | 22.50 | 6.50 | 3.20 | 0.52 | 86.31 | 22.42 |
| | BDMD* | 20.38 | 20.40 | 21.30 | 16.60 | 19.80 | 21.80 | 6.80 | 2.10 | 1.10 | 13.58 | 14.39 |
| | EPS-AD | 85.71 | **100.00** | 87.50 | 99.90 | 72.80 | **99.90** | 10.10 | 14.90 | 11.53 | **99.90** | 68.22 |
| | EPS-AD* | 97.58 | 100.00 | 98.80 | 100.00 | 81.10 | 99.90 | 12.90 | 17.50 | 15.05 | 100.00 | 72.28 |
| | PID | **90.02** | 91.80 | **92.80** | 94.10 | **88.70** | 88.50 | 89.80 | 87.60 | 89.17 | 85.19 | **89.77** |
| ADV | FS | 0.95 | 9.80 | 0.75 | 17.55 | 16.18 | 24.56 | 10.20 | 4.68 | 14.89 | 4.11 | 10.37 |
| | DiffPure | 1.90 | 35.36 | 1.50 | 46.99 | 7.12 | 16.27 | 22.40 | 4.52 | 12.58 | 4.11 | 15.28 |
| | BDMD | 8.57 | 8.87 | 8.27 | 8.60 | 6.54 | 9.19 | 7.60 | 8.62 | 12.14 | 2.70 | 8.03 |
| | BDMD* | 3.81 | 15.23 | 3.76 | 11.67 | 5.23 | 10.47 | 10.40 | 5.92 | 19.40 | 4.05 | 8.99 |
| | EPS-AD | 50.48 | **100.00** | 47.37 | **100.00** | 89.00 | 99.86 | 86.10 | 8.87 | 68.51 | **100.00** | 75.02 |
| | EPS-AD* | 76.19 | 100.00 | 74.44 | 100.00 | 92.23 | 100.00 | 90.50 | 7.42 | 74.65 | 100.00 | 81.54 |
| | PID | **77.14** | 80.26 | **76.69** | 82.09 | 81.88 | 80.62 | 89.70 | 80.65 | 90.85 | 76.71 | **81.66** |

when the detector is trained and tested on AEs generated by the same attack against the same primal model. Specifically, for each attack, the corresponding AEs are split into 60% for training, 10% for validation, and 30% for testing. The detector is trained for 100 epochs with a batch size of 80 and a learning rate of 0.001. The AUC score is then computed on the test set for each attack.

For the black-box SID, we use the naturally and adversarially trained VGG16 models as the primal models to mitigate performance degradation caused by the architectural differences from VGG19, ensuring a competitive baseline. The naturally trained VGG16 model achieves 92.19% test accuracy, and the adversarially trained one, trained using PGD-8/255, achieves 79.29% test accuracy. The dual classifier composed of the same VGG16 architecture and a WAWT layer is also trained. AEs are then generated by attacks against these two VGG16 models, with each attack configured using the same parameters as described in Section A.2. The AE split and training parameters are the same as previously described to train the detector. When detecting AEs generated against the naturally and adversarially trained VGG19 models, we use a detector trained on AEs from the same attack, but generated against the naturally and adversarially trained VGG16 models, respectively.

**Expected Perturbation Score-based Adversarial Detection (EPS-AD) (White-Box Detection).** We adopt the EPS-AD as the detection baseline on ImageNet. The implementation follows the released code[9], where the pre-trained Guided Diffusion (Dhariwal & Nichol, 2021) is adopted to estimate the expected perturbation score (EPS) of test samples and timestep $t^* = 0.05$. Then, the EPS-based maximum mean discrepancy (MMD) is used as the metric to measure the discrepancy between NEs and AEs and train the detector, where the detector structure is the same as the one described in Zhang et al. (2023).

For the white-box EPS-AD (denoted as EPS-AD*), we generate 10,000 $\ell_\infty$-FGSM and $\ell_2$-FGSM AEs with a perturbation size of 1/255, along with 10,000 NEs to calculate their EPSs and train the detector. When detecting AEs generated against the naturally and adversarially trained ResNet50 models, we use detectors trained using FGSM AEs generated against the same models, respectively. Detectors are trained for 200 epochs with a batch size of 200 and a learning rate of 0.002.

For the black-box EPS-AD, when detecting AEs generated against the naturally and adversarially trained ResNet50 models, we use detectors trained on FGSM AEs generated against the naturally and adversarially trained ResNet101 models, respectively. For the two ResNet101 models, we use pre-trained weights under natural[1] and adversarial[2] training settings, respectively. The number of AEs and NEs, along with the training settings, are consistent with those described above.

---

[9]https://github.com/ZSHsh98/EPS-AD

Table 10: AUC scores (%) of PID on ImageNet, where mean and standard deviation are calculated across three random subsets, and NAT (ADV) means the primal model is naturally (adversarially) trained.

| Primal Model | PGD $\epsilon = \frac{1}{255}$ | PGD $\epsilon = \frac{8}{255}$ | AA $\epsilon = \frac{1}{255}$ | AA $\epsilon = \frac{8}{255}$ | C&W $\kappa = 0$ | C&W $\kappa = 1$ | DeepFool | Square | TA | VNI-FGSM |
|---|---|---|---|---|---|---|---|---|---|---|
| NAT | $98.19 \pm 0.27$ | $98.46 \pm 0.34$ | $98.64 \pm 0.26$ | $98.90 \pm 0.60$ | $98.61 \pm 0.32$ | $98.26 \pm 0.05$ | $98.13 \pm 0.35$ | $97.89 \pm 0.35$ | $97.78 \pm 0.04$ | $98.42 \pm 0.29$ |
| ADV | $95.83 \pm 0.13$ | $96.26 \pm 0.51$ | $98.34 \pm 0.27$ | $97.65 \pm 0.14$ | $96.96 \pm 0.20$ | $97.34 \pm 0.36$ | $98.37 \pm 0.23$ | $97.21 \pm 0.32$ | $98.41 \pm 0.18$ | $92.57 \pm 0.27$ |

Table 11: Detection performance of PID on PGD vs. three adaptive attacks on CIFAR-10 and ImageNet, where NAT (ADV) means the primal model is naturally (adversarially) trained.

| Dataset | Primal Model | Auxiliary Model | PGD ASR | PGD AUC | A1 ASR | A1 AUC | A2 ASR | A2 AUC | A3 ASR | A3 AUC |
|---|---|---|---|---|---|---|---|---|---|---|
| CIFAR-10 | NAT | ResNet34_ADV | 100.00 | 96.56 | 98.96 | 96.94 | 84.75 | 95.86 | 99.89 | 91.92 |
| | | ViT-L/16 | | 98.54 | 92.70 | 3.83 | 92.61 | 2.56 | 99.89 | 72.20 |
| | | CLIP-ViT-L/14 | | 97.55 | 24.52 | 97.94 | 13.72 | 98.23 | 99.25 | 33.44 |
| | ADV | ResNet34_ADV | 45.37 | 91.46 | 16.66 | 89.02 | 7.97 | 88.60 | 20.20 | 76.43 |
| | | ViT-L/16 | | 99.68 | 20.68 | 84.58 | 19.77 | 54.18 | 36.84 | 51.82 |
| | | CLIP-ViT-L/14 | | 98.75 | 11.87 | 98.84 | 8.04 | 98.55 | 3.48 | 16.27 |
| ImageNet | NAT | ConvNeXt-S_ADV | 100.00 | 97.53 | 100.00 | 98.37 | 100.00 | 98.36 | 100.00 | 84.43 |
| | | ViT-L/16 | | 98.54 | 100.00 | 3.41 | 100.00 | 3.35 | 100.00 | 27.39 |
| | | CLIP-ViT-L/14 | | 95.91 | 87.70 | 88.86 | 39.60 | 90.41 | 98.70 | 16.18 |
| | ADV | ConvNeXt-S_ADV | 75.50 | 97.26 | 21.80 | 94.28 | 11.80 | 94.50 | 62.10 | 65.93 |
| | | ViT-L/16 | | 96.17 | 22.70 | 63.63 | 22.70 | 61.02 | 26.30 | 23.60 |
| | | CLIP-ViT-L/14 | | 96.13 | 24.30 | 91.97 | 1.70 | 94.29 | 8.40 | 8.04 |

## A.4 DETECTION PERFORMANCE EVALUATED BY TPR AT 5% FPR

To comprehensively evaluate PID and the baselines, we fix the FPR at 5% and report the corresponding TPR for each detection method. The results on CIFAR-10 and ImageNet are presented in Table 8 and Table 9, respectively. The results show that PID consistently achieves high TPRs. For example, on CIFAR-10 it reaches 97.45% and 97.25% when the primal models are naturally trained and adversarially trained, respectively. On ImageNet, EPS-AD attains comparable TPRs, but its performance drops significantly against score-based and decision-based black-box attacks. For example, its TPR falls to only 14.90% and 11.53% when detecting Square and TA attacks on the naturally trained model.

## A.5 ADDITIONAL EXPERIMENTS ON RANDOM IMAGE SAMPLES ON IMAGENET

For evaluation on ImageNet, we randomly select 1000 images (one from each class) to assess the detector. To mitigate the effect of randomness and ensure the robustness of the results, we additionally sample two separate sets of 1000 images each for both naturally and adversarially trained model. The mean AUC scores and standard deviations across these three subsets are reported in Table 10, showing that PID remains stable under different random splits.

## A.6 ADAPTIVE ATTACKS

We design three adaptive attacks to evaluate the robustness of our PID. In addition to the strongest attack described in Section 4.3, the other two are presented here. Specifically, increasing $g_y(x)$ in Metric 1 used in PID can also be achieved by performing a targeted attack against two models, forcing both primal and auxiliary models to misclassify the AE into the same label. This can be expressed as,

$$\min_r \mathcal{L}(f(x+r), t) + \lambda \mathcal{L}(g(x+r), t) \quad \text{s.t.} \ \|r\|_\infty \le \epsilon, \qquad (8)$$

where $t$ is the targeted label satisfying $t \neq y_{true}$, $y_{true}$ is the ground-truth label of $x$, $\mathcal{L}(\cdot, \cdot)$ is the loss function, $\epsilon$ is the perturbation constraint of the perturbation $r$, and $\lambda$ is a trade-off parameter.

We employ the PGD attack strategy to generate the perturbation $r$, with the perturbation budget constrained to $\epsilon = 8/255$, a total of 100 iterations, and a step size of $1/255$. The adaptive attack

Table 12: Comparison of AUC scores (%) of PID by employing different metrics for quantifying the prediction inconsistency on CIFAR-10, where NAT (ADV) means the primal model is naturally (adversarially) trained.

| Primal Model | Metric | PGD $\epsilon=\frac{1}{255}$ | PGD $\epsilon=\frac{8}{255}$ | AA $\epsilon=\frac{1}{255}$ | AA $\epsilon=\frac{8}{255}$ | C&W $\kappa=0$ | C&W $\kappa=1$ | DeepFool | Square | TA | VNI-FGSM | Average |
|---|---|---|---|---|---|---|---|---|---|---|---|---|
| NAT | $I_{pred}=1-g_y(x)$ | **99.81** | 98.54 | **99.85** | 98.71 | **99.93** | **99.45** | **99.86** | **99.88** | **99.85** | 97.02 | **99.29** |
| | $I_{pred}=f_y(x)-g_y(x)$ | 99.59 | **98.80** | 99.68 | **98.94** | 99.52 | 97.81 | 99.48 | 97.98 | 96.70 | **97.34** | 98.58 |
| | $I_{pred}=\|f(x)_{\text{top}-n}-g(x)_{\text{top}-n}\|_1$ | 99.44 | 97.58 | 99.53 | 97.82 | 99.67 | 99.08 | 99.62 | 99.46 | 99.40 | 95.94 | 98.75 |
| | $I_{pred}=\|f(x)-g(x)\|_1$ | 99.58 | 98.20 | 99.65 | 98.42 | 99.71 | 99.21 | 99.75 | 99.37 | 99.30 | 96.38 | 98.96 |
| ADV | $I_{pred}=1-g_y(x)$ | **99.87** | **99.68** | **99.88** | **99.69** | **99.91** | **99.90** | **99.55** | **99.82** | **99.82** | **94.90** | **99.30** |
| | $I_{pred}=f_y(x)-g_y(x)$ | 99.34 | 97.74 | 99.37 | 97.60 | 98.30 | 98.24 | 92.23 | 98.12 | 95.94 | 90.48 | 96.74 |
| | $I_{pred}=\|f(x)_{\text{top}-n}-g(x)_{\text{top}-n}\|_1$ | 71.50 | 73.35 | 69.83 | 74.37 | 89.58 | 92.12 | 87.36 | 91.04 | 82.73 | 45.71 | 77.76 |
| | $I_{pred}=\|f(x)-g(x)\|_1$ | 66.08 | 91.16 | 63.30 | 90.58 | 84.73 | 88.49 | 86.77 | 89.60 | 76.63 | 52.06 | 78.94 |

Table 13: Comparison of AUC scores (%) of PID by employing different models as the auxiliary model on CIFAR-10, where NAT (ADV) means the primal model is naturally (adversarially) trained.

| Primal Model | Auxiliary Model | PGD $\epsilon=\frac{1}{255}$ | PGD $\epsilon=\frac{8}{255}$ | AA $\epsilon=\frac{1}{255}$ | AA $\epsilon=\frac{8}{255}$ | C&W $\kappa=0$ | C&W $\kappa=1$ | DeepFool | Square | TA | VNI-FGSM | Average |
|---|---|---|---|---|---|---|---|---|---|---|---|---|
| NAT | ResNet34_ADV | 88.18 | 96.56 | 90.25 | 96.70 | 96.93 | 96.68 | 96.96 | 96.74 | 96.46 | 95.93 | 95.14 |
| | ViT-L/16 | **99.81** | **98.54** | **99.85** | **98.71** | **99.93** | **99.45** | **99.86** | **99.88** | **99.85** | **97.02** | **99.29** |
| | CLIP-ViT-L/14 | 98.53 | 97.55 | 98.73 | 97.95 | 99.25 | 98.40 | 99.08 | 99.22 | 99.31 | 95.57 | 98.36 |
| ADV | ResNet34_ADV | 86.98 | 91.46 | 87.18 | 92.12 | 97.52 | 96.01 | 97.18 | 95.06 | 97.51 | 86.47 | 92.75 |
| | ViT-L/16 | **99.87** | **99.68** | **99.88** | **99.69** | **99.91** | **99.90** | **99.55** | **99.82** | **99.82** | **94.90** | **99.30** |
| | CLIP-ViT-L/14 | 98.86 | 98.75 | 99.12 | 98.69 | 99.28 | 99.18 | 98.13 | 98.97 | 99.33 | 93.43 | 98.37 |

is denoted as A1 when $\lambda=1$ in Eq. 8, and as A2 when $\lambda=99$, which places more emphasis on fooling the auxiliary model. The two-phase adaptive attack described in Section 4.3 is denoted as A3. The performance of PID against these three adaptive attacks on CIFAR-10 and ImageNet is reported in Table 11, where all three kinds of auxiliary models are considered.

As shown in Table 11, among the three adaptive attacks, A3 proves to be the most effective against PID, yielding high ASRs and low AUC scores. On both datasets, although A2 is highly effective against PID when the auxiliary model is ViT-L/16, its effectiveness drops substantially when the auxiliary model is CLIP-ViT-L/14, which may be attributed to the increased difficulty of jointly attacking CNN and CLIP models. Overall, when adversarially trained CNNs are used as auxiliary models, PID remains robust: its AUC drops by less than 3% under A1 and A2, and although A3 reduces AUC further, PID still maintains satisfactory detection performance.

### A.7 FULL EXPERIMENTAL RESULTS ON METRICS FOR QUANTIFYING PREDICTION INCONSISTENCY

The detection performance of PID by employing different metrics for quantifying the prediction inconsistency against each attack on CIFAR-10 is summarized in Table 12. It can be observed that using $I_{pred}=1-g_y(x)$ helps the PID to achieve the best detection performance in most cases. After the primal model is adversarially trained, the effectiveness of metrics $I_{pred}=\|f(x)_{\text{top}-n}-g(x)_{\text{top}-n}\|_1$ and $I_{pred}=\|f(x)-g(x)\|_1$ vary according to strengths and types of attacks, generally exhibiting a significant decline.

### A.8 FULL EXPERIMENTAL RESULTS ON CHOICE OF AUXILIARY MODELS

The detection performance of PID by employing different auxiliary models against each attack on CIFAR-10 and ImageNet is summarized in Table 13 and Table 14, respectively. It can be observed that ViT-L/16 consistently achieves the highest AUC scores on CIFAR-10, regardless of whether it is combined with naturally or adversarially trained models. While on ImageNet, especially after the primal model is adversarially trained, ConvNeXt-S_ADV can sometimes achieve better detection performance, which is related to its superior performance and robustness compared with traditional CNNs. In addition, the CLIP model exhibits lower accuracy on NEs than the other two models, leading to a higher FPR on NEs and degrading detection performance. Nevertheless, pre-trained

Table 14: Comparison of AUC scores (%) of PID by employing different models as the auxiliary model on ImageNet, where NAT (ADV) means the primal model is naturally (adversarially) trained.

| Primal Model | Auxiliary Model | PGD $\epsilon=\frac{1}{255}$ | PGD $\epsilon=\frac{8}{255}$ | AA $\epsilon=\frac{1}{255}$ | AA $\epsilon=\frac{8}{255}$ | CW $\kappa=0$ | CW $\kappa=1$ | DeepFool | Square | TA | VNI-FGSM | Average |
|---|---|---|---|---|---|---|---|---|---|---|---|---|
| NAT | ConvNeXt-S_ADV | 96.33 | 97.53 | 97.06 | 98.00 | 95.82 | 96.32 | 96.24 | 96.48 | 96.81 | **98.70** | 96.93 |
| | ViT-L/16 | **98.45** | **98.54** | **98.74** | **98.90** | **98.17** | **98.20** | **98.36** | 97.80 | 97.80 | 98.11 | **98.31** |
| | CLIP-ViT-L/14 | 95.15 | 95.91 | 95.83 | 96.95 | 93.94 | 94.47 | 95.37 | 94.61 | 95.86 | 95.29 | 95.34 |
| ADV | ConvNeXt-S_ADV | 87.17 | **97.26** | 88.82 | 97.57 | 94.77 | 96.94 | **98.54** | 96.45 | **99.03** | **94.03** | 95.06 |
| | ViT-L/16 | **95.98** | 96.17 | **98.20** | 97.83 | **97.23** | **97.11** | 98.06 | **96.90** | 98.34 | 92.23 | **96.81** |
| | CLIP-ViT-L/14 | 90.98 | 96.13 | 91.52 | 96.07 | 93.55 | 94.33 | 95.98 | 93.40 | 97.42 | 93.39 | 94.28 |

Table 15: Comparison of GPU memory and inference time for PID with three auxiliary models and baselines on ImageNet.

| Computational Cost | FS | DiffPure | BDMD | EPS-AD | PID (ConvNext-S_ADV) | PID (ViT-L/16) | PID (CLIP-ViT-L/14) |
|---|---|---|---|---|---|---|---|
| GPU Memory/MiB | 2138 | 28346 | 3568 | 40030 | 2444 | 3284 | 5690 |
| Inference Time (per batch)/ms | 2171.27 | 142881.58 | 1260.62 | 39154.30 | 76.33 | 288.81 | 645.26 |

foundation models like CLIP are readily accessible and require no task-specific training, making them appealing choices for deploying PID under more constrained detection scenarios, such as scenarios where the training set is completely unknown.

### A.9 COMPUTATIONAL COSTS OF PID AND BASELINES ON IMAGENET

Experiments are conducted using PyTorch on two NVIDIA A800 80GB PCIe GPUs. A detailed comparison of the computational cost of PID with three auxiliary models and other baselines on ImageNet is provided in Table 15, with the batch size fixed at 16. The results show that (1) PID is lightweight. Although it uses slightly more GPU memory than FS, it achieves faster inference across all auxiliary models. (2) Larger auxiliary models lead to higher memory usage, but the overall cost is still much lower than that of DiffPure and EPS-AD.

### A.10 MORE CHOICES OF AUXILIARY MODELS IN PID ON IMAGENET

In addition to the three auxiliary models already evaluated before, we have further selected four additional models, namely the adversarially trained ConvNeXt-L[2] (denoted as ConvNeXt-L_ADV), the pre-trained ViT-B/16[3], the pre-trained ViT-H/14[3], and the pre-trained CLIP-ViT-B/16[10].

We conduct additional experiments on ImageNet using these auxiliary models, which can be found in Table 16. The results demonstrate that, as long as the auxiliary model differs from the primal model either in training approach or architecture, PID remains effective. Moreover, it can be observed that larger models generally yield better detection performance when sharing the same architecture. For example, ConvNeXt-L_ADV achieves AUC scores of 97.68% and 96.89% when detecting AEs generated against the naturally and adversarially trained primal models, respectively, outperforming ConvNeXt-S_ADV, which achieves 96.93% and 95.06% in the same settings. We attribute this phenomenon to the fact that larger models (differ from the primal model either in training approach or architecture) tend to achieve higher accuracy on NEs, resulting in more consistency between the primal and auxiliary models when classifying NEs, and more pronounced inconsistency when classifying AEs.

Overall, the results in Table 16 consistently demonstrate that PID remains generalizable across diverse auxiliary architectures. Specifically, **6 out of 7** auxiliary models outperform the second-best result in Table 3, both when detecting AEs against the naturally and adversarially trained primal models.

---

[10]https://huggingface.co/openai/https://huggingface.co/openai/clip-vit-base-patch16

Table 16: Comparison of AUC scores (%) of PID by employing more auxiliary models on ImageNet, where NAT (ADV) means the primal model is naturally (adversarially) trained.

| Primal Model | Detection Method | PGD $\epsilon=\frac{1}{255}$ | PGD $\epsilon=\frac{8}{255}$ | AA $\epsilon=\frac{1}{255}$ | AA $\epsilon=\frac{8}{255}$ | C&W $\kappa=0$ | C&W $\kappa=1$ | DeepFool | Square | TA | VNI-FGSM | Average |
|---|---|---|---|---|---|---|---|---|---|---|---|---|
| NAT | ConvNext-S_ADV | 96.33 | 97.53 | 97.06 | 98.00 | 95.82 | 96.32 | 96.24 | 96.48 | 96.81 | 98.70 | 96.93 |
| | ConvNext-L_ADV | 97.26 | 98.07 | 97.78 | 98.53 | 96.80 | 97.31 | 97.10 | 97.21 | 97.59 | 99.13 | 97.68 |
| | ViT-B/16 | 98.34 | 98.25 | 98.66 | 98.55 | 98.01 | 98.06 | 98.12 | 97.54 | 97.58 | 96.85 | 98.00 |
| | ViT-L/16 | 98.45 | 98.54 | 98.74 | 98.90 | 98.17 | 98.20 | 98.36 | 97.80 | 97.80 | 98.11 | 98.31 |
| | ViT-H/14 | 99.12 | 99.18 | 99.39 | 99.36 | 98.78 | 98.98 | 99.18 | 98.49 | 98.31 | 98.39 | 98.92 |
| | CLIP-ViT-B/16 | 91.48 | 92.30 | 92.83 | 93.65 | 90.15 | 90.42 | 92.14 | 90.93 | 93.16 | 90.63 | 91.77 |
| | CLIP-ViT-L/14 | 95.15 | 95.91 | 95.83 | 96.95 | 93.94 | 94.47 | 95.37 | 94.61 | 95.86 | 95.29 | 95.34 |
| ADV | ConvNext-S_ADV | 87.17 | 97.26 | 88.82 | 97.57 | 94.77 | 96.94 | 98.54 | 96.45 | 99.03 | 94.03 | 95.06 |
| | ConvNext-L_ADV | 92.38 | 98.27 | 92.39 | 98.49 | 96.33 | 98.23 | 99.11 | 96.90 | 99.39 | 97.37 | 96.89 |
| | ViT-B/16 | 95.86 | 96.02 | 97.72 | 97.66 | 97.01 | 96.94 | 97.78 | 96.64 | 98.18 | 92.01 | 96.58 |
| | ViT-L/16 | 95.98 | 96.17 | 98.20 | 97.83 | 97.23 | 97.11 | 98.06 | 96.90 | 98.34 | 92.23 | 96.81 |
| | ViT-H/14 | 97.04 | 97.28 | 99.13 | 98.90 | 98.26 | 98.15 | 99.02 | 97.92 | 99.24 | 93.81 | 97.88 |
| | CLIP-ViT-B/16 | 87.57 | 92.33 | 87.18 | 92.53 | 90.74 | 90.37 | 92.87 | 89.38 | 95.53 | 88.80 | 90.73 |
| | CLIP-ViT-L/14 | 90.98 | 96.13 | 91.52 | 96.07 | 93.55 | 94.33 | 95.98 | 93.40 | 97.42 | 93.39 | 94.28 |

