# OpenReview forum: "Prediction Inconsistency Helps Generalizable Detection of Adversarial Examples"
_ICLR.cc/2026/Conference — ICLR 2026 Conference Withdrawn Submission_

### Official Review · Reviewer_K8nF · 2025-10-28

**Soundness:** 1
**Presentation:** 2
**Contribution:** 1
**Rating:** 2
**Confidence:** 3

**Summary:**

This paper proposes a new detection technique for adversarial examples that calculates metrics based on the prediction inconsistency between a primal model and an auxiliary model. A threshold based on the metric is used to identity adversarial examples. Through experiments across many different attack algorithms and by comparing with a number of existing detection methods, they show the effectiveness of their technique, and present the influence of many factors like choice of auxiliary model.

**Strengths:**

1. The goal of a lighter weight and easy to implement adversarial detection method is important given that many methods are computationally expensive or require training custom detection classifiers.

2. The method has the advantage of being very simple, merely making use of the confidence score of the auxiliary model on the predicted label.

3. The authors comprehensively study attack types, and provide analysis for numerous aspects like the influence of the choice of auxiliary model and adaptive attacks.

**Weaknesses:**

1. The paper claims the superiority of PID compared to SOTA detection methods but my main concern is that the method would have a high false positive rate for misclassifying clean examples since for the main metric it is just using the confidence score of the auxiliary classifier on the predicted label of the primal, so any large discrepancies in confidence between the primal and auxiliary model would in theory be taken as an adversarial example. Furthermore, the method does not consider that the primal classifier could be outputting a high entropy prediction because it is a genuine hard example (and likewise the auxiliary too). I could easily see a lot of situations where both models have low confidence scores on normal examples but metric 1 is close to 1. This calls into question model calibration and how this affects PID and its detection performance and misclassifications.
There are some results in the appendix (Tables 8 and 9) at a fixed false positive rate but there is no comparison of the overall rate the PID would classify clean samples as adversarial to my knowledge.

2. There is a lack of details or discussion about setting the detection threshold in the paper.

3. The paper lacks deeper analysis on why the method is effective. Previous methods like BDMD and SID already exploit prediction inconsistency and train classifiers on top of the features of both clean and adversarial examples. The main difference with this paper is that they forgo training a detector and just use the labels/confidence directly and detect based on a set threshold. Particularly in the past work SID, they provide extensive theoretical analysis on forming the dual classifier, including how to create a favorable transformed decision boundary. In comparison this paper claims that just dropping the trained detector and using the confidence of an adversarially trained auxiliary model leads to significantly better performance (regardless of the auxiliary model used, detection AUC of PID is close to or above 90%). Nonetheless, metric 1 doesn't even truly require the primal model. You could just use the auxiliary model directly, since based on lines 198-199 if we assume it does fall for an attack then the confidence will be low anyway and you can just set a cutoff for low confidence predictions. I believe the method needs to be better motivated and explained how the purported benefits are being achieved.

4.  Multiple detection methods are only tested on one dataset (SID only on CIFAR and EPS-AD only on ImageNet) which makes it hard to gauge the effectiveness of these methods compared to PID.

5. The method is stated to be notably weaker against adaptive attacks particularly when ViT or CLIP are the auxiliary models, and there is no comparison to how other detection methods fare with them (i.e., if those other methods are superior against them). Since adaptive attacks are a strong indicator of the performance of a detection method and need to be considered in practical deployment, to fully motivate adopting this method requires this comparison.

6. I observe a lot of cases where detection performance is higher on more subtle attacks with a lower $\epsilon$ such as in Table 2, which is a bit paradoxical.

**Questions:**

Can a discussion and be provided on why forgoing a classifier and merely using a set threshold appears to lead to far better and more consistent performance compared to the alternative more theoretically informed methods?

Can additional analysis be provided on whether the method has a high rate of classifying normal examples as adversarial?

How is the threshold set, and what would be the process for optimizing it in practical use?

I think given that confidence is crucial in PID, a calibration analysis would be helpful to more concretely see how choice of models affects performance. How does model primal or auxiliary model miscalibration is correlated with detection accuracy?

---

### Official Review · Reviewer_hg9G · 2025-10-28

**Soundness:** 2
**Presentation:** 2
**Contribution:** 1
**Rating:** 2
**Confidence:** 4

**Summary:**

The paper presents a defense against adversarial examples by checking the disagreement between the outputs of the defended model and an auxiliary model. The authors propose four different metrics to quantify this disagreement and evaluate the defense against several attacks and baseline defenses. The best performing metric is computed as the complementary output probability of the auxiliary model with respect to the class predicted by the defended model, which can be used as a rejection criterion after being thresholded.

**Strengths:**

- The paper proposes a simple solution for a relevant and still challenging problem.

**Weaknesses:**

- Novelty. This defense falls in the category of ensemble-based approaches, which includes several works that follow similar procedures to achieve adversarial robustness [a-e]. With respect to them, I believe this paper does not propose novel ideas.
- Technical contribution. The provided methodology simply checks how much adversarial examples transfer to another model. From a technical standpoint, this contribution appears weak.
- Experimental results and evaluation. The main experimental results refer to a setting where the attacker can access the defended model only. This does not follow the best practices on evaluating adversarial defenses, as it might overestimate its performance, whereas the robustness evaluations should be performed against worst-case scenarios with the strongest attacker. Additionally, I feel like the defense might be easily circumvented by leveraging state-of-the-art transfer-based attacks (see https://github.com/Trustworthy-AI-Group/TransferAttack), especially if the attacker can access both the models (in this case, it is likely that the simple ensemble attack [f] is enough to fool both of them and bypass the defense). On the other hand, the proposed adaptive attacks seem to be suboptimal.

[a] Pang, T., Xu, K., Du, C., Chen, N., & Zhu, J. (2019). Improving Adversarial Robustness via Promoting Ensemble Diversity. ArXiv, abs/1901.08846.

[b] Wei, W., & Liu, L. (2020). Robust Deep Learning Ensemble Against Deception. IEEE Transactions on Dependable and Secure Computing, 18, 1513-1527.

[c] Bui, T., Le, T., Zhao, H., Montague, P., deVel, O., Abraham, T., & Phung, D.Q. (2020). Improving Ensemble Robustness by Collaboratively Promoting and Demoting Adversarial Robustness. AAAI Conference on Artificial Intelligence.

[d] Huang, B., Ke, Z., Wang, Y.A., Wang, W., Shen, L., & Liu, F. (2021). Adversarial Defence by Diversified Simultaneous Training of Deep Ensembles. AAAI Conference on Artificial Intelligence.

[e] Feldsar, B., Mayer, R., & Rauber, A. (2023). Detecting Adversarial Examples Using Surrogate Models. Mach. Learn. Knowl. Extr., 5, 1796-1825.

[f] Liu, Y., Chen, X., Liu, C., & Song, D.X. (2016). Delving into Transferable Adversarial Examples and Black-box Attacks. ArXiv, abs/1611.02770.

**Questions:**

- In the paper, the authors distinguish between black- and white-box defense methods. I felt confused, as this use of that terminology is not usual in the literature. Could you please explain its meaning?

---

### Official Review · Reviewer_1u7z · 2025-10-31

**Soundness:** 2
**Presentation:** 3
**Contribution:** 2
**Rating:** 4
**Confidence:** 3

**Summary:**

This paper proposes a detection method called Prediction Inconsistency Detection (PID), which leverages the discrepancy between a primal model and an auxiliary model to identify adversarial examples (AEs). The main intuition is that AEs often cause inconsistent predictions between models trained differently (e.g., different architectures or training regimes), while normal examples (NEs) yield consistent outputs. The authors design several metrics to quantify prediction inconsistency, select the simplest one (based on hard-label confidence), and test PID across multiple datasets (CIFAR-10, ImageNet) and attack types (white-box, black-box, mixed). They report high AUCs and claim superior generalization and robustness compared to five state-of-the-art detectors.

**Strengths:**

1. The concept of using model disagreement as a signal for adversarial detection is straightforward and interpretable. It requires no extra detector training, making it lightweight.

2. The paper includes experiments on both CIFAR-10 and ImageNet, multiple attack categories (white-box, black-box, mixed, and adaptive), and compares with well-known baselines (FS, DiffPure, BDMD, SID, EPS-AD).

3. The paper is clearly structured, with detailed experimental settings and appendices that improve reproducibility.

**Weaknesses:**

1. The idea is closely related to prior works such as Bi-model Decision Mismatch (BDMD) and Sensitivity Inconsistency Detector (SID), which also leverage model disagreement. The difference is primarily that PID avoids explicit detector training. This simplification, while practical, does not constitute a strong conceptual advance over existing inconsistency-based detectors.

2. The main claim—that prediction inconsistency is a reliable, generalizable signal for adversarial detection—is purely empirical. The paper does not provide a theoretical explanation of why prediction inconsistency correlates with adversarial perturbations or how it scales with model diversity, dataset complexity, or attack type.

3. PID is claimed to be a “black-box detection” method, but it assumes access to the primal model’s predictions (and sometimes labels), which implies partial white-box knowledge. The term “black-box” is therefore misleading. True black-box detectors should not rely on internal confidence values or consistent prediction pairs.

4. The “adaptive” attacks described in Section 4.3 are simplistic and limited to PGD-style optimization. They do not simulate realistic adaptive adversaries such as [1,2] that directly minimize detection scores. PID’s robustness under fully adaptive, detection-aware scenarios remains unconvincing.

[1] Reliable Evaluation of Adversarial Robustness with an Ensemble of Diverse Parameter-free Attacks, ICML 2020.
[2] On Adaptive Attacks to Adversarial Example Defenses, NeurIPS 2020.

**Questions:**

1. How does PID perform against a fully adaptive attacker that directly minimizes the inconsistency score $I_{\textrm{pred}}$?

2. Is PID provably more difficult to bypass than other inconsistency-based detectors?

---

### Note · Authors · 2025-12-04

I have read and agree with the venue's withdrawal policy on behalf of myself and my co-authors.